# Italians Can Resist Everything, Except Flat-Faced Dogs! [note 1]

**DOI:** 10.3390/ani15101496

**Published:** 2025-05-21

**Authors:** Simona Cannas, Clara Palestrini, Sara Boero, Alice Garegnani, Silvia M. Mazzola, Emanuela Prato-Previde, Greta V. Berteselli

**Affiliations:** 1Department of Veterinary Medicine and Animal Science, University of Milan, 26900 Lodi, Italyclara.palestrini@unimi.it (C.P.); sara.boero@unimi.it (S.B.); alice.garegnani@unimi.it (A.G.); silvia.mazzola@unimi.it (S.M.M.); 2Department of Pathophysiology and Transplantation, University of Milan, 20133 Milan, Italy; emanuela.pratoprevide@unimi.it

**Keywords:** brachycephalic breed, dog behaviors, human–animal bond, owner motivations, pet ownership, welfare

## Abstract

Exaggerated anatomical features in dogs, particularly a short nose, broad face, and round eyes typical of breeds such as the French bulldog, English bulldog, and pug, are increasingly popular. However, concern about the welfare of these breeds is growing due to the health problems associated with their facial structure. Understanding owners’ motivations and satisfaction, their perception of their dog’s health status, the quality of the human–dog relationship, and the dog’s behavioral traits can provide valuable insights to improve animal welfare in a long-term and meaningful way. Our results show that the French bulldog is the most popular among brachycephalic dog breeds. Despite being aware of breed-related issues and the presence of clinical problems, most owners reported being satisfied with their dogs, and rated their health status as “good”. Furthermore, these breeds were primarily chosen for their personality and aesthetic appeal. Owning a flat-faced dog is a complex and multidimensional phenomenon. A combination of motivations, misconceptions, and emotional engagement plays a significant role in sustaining this trend.

## 1. Introduction

The popularity of flat-faced dogs is rising worldwide, despite the numerous and serious health problems they often experience. This phenomenon, known as the *brachycephalic paradox*, has led to a welfare crisis for these breeds [1]. Selective breeding for extreme facial morphology has increased the frequency of breed-related disorders. Among brachycephalic breeds, English and French bulldogs, pugs, and Boston terriers are the most affected due to their hypertypic conformation [2].

Brachycephaly results from a genetic mutation that alters the development of the basioccipital and basisphenoid bones, leading to a shortening of the basicranial axis. This reduction in the length of the facial skeleton is not accompanied by proportional shortening of the soft tissues (e.g., soft palate, tongue, and tonsils) [2,3]. These disproportionately large soft tissues increase airway resistance and air turbulence, potentially causing secondary changes such as soft palate and laryngeal edema, swelling, saccule and tonsil eversion, and even laryngeal collapse. These clinical features, along with other anomalies such as tracheal hypoplasia and aberrant turbinates, contribute to life-threatening respiratory compromise. Collectively, these issues define Brachycephalic Obstructive Airway Syndrome (BOAS) [3,4,5]. Moreover, respiratory distress is often accompanied by difficulties in ingesting food and uncoordinated swallowing, leading to aerophagia and aspiration, which can result in gastric tympany and distention, along with increased gastrin secretion [6]. Brachycephalic dogs frequently present with gastrointestinal signs, including reflux and vomiting, rather than with respiratory problems. Chronic gastritis and lymphoplasmacytic duodenitis are also commonly associated with BOAS [3,6,7,8]. A distinct brachycephalic ocular syndrome (BOS) has been identified. The typical conformational features of flat-faced dogs, such as shallow orbits and large eyelid openings, contribute to abnormally protruding eyes, increasing the risk of ocular conditions such as corneal ulcers [9,10,11]. In addition, brachycephalic breeds are prone to other clinical issues, such as dystocia in dams, heat intolerance, and neurological and dermatological disorders [12,13]. The severity of these health problems significantly affects the welfare of brachycephalic dogs and can also impact human well-being (i.e., caregiver burden) [14,15,16]. Nevertheless, these concerns do not appear to deter owners from acquiring or reacquiring brachycephalic dogs [17]. In many cases, emotional attachment and aesthetic appeal (e.g., the “cuteness effect” or baby schema) take precedence over concerns about health, quality of life, or longevity [1,18,19,20]. Signs of poor health are often normalized in these breeds. This normalization, combined with misconceptions regarding positive behavioral traits of brachycephalic breeds (i.e., good for children, good companion) could lead to positive perceptions of these breeds, feeding the desire to continue owning the breed [1,20,21,22,23]. Furthermore, trends in fashion, social media, and advertising can influence decision-making when acquiring a specific breed [20,24,25]. This complex scenario highlights the need for a multidisciplinary and evidence-based approach. Simple educational messages and awareness campaigns have not been sufficient to address the brachycephalic crisis effectively [26,27]. Multidisciplinary and scientific insights are needed to better understand this multifaceted phenomenon and to guide owners and breeders toward more ethical choices, including the revision of breed standards to improve animal welfare in a long-term and meaningful way. In light of this, the present study aimed to explore the brachycephalic phenomenon in a sample of Italian dog owners. Specifically, it investigated differences between brachycephalic dog owners (BDOs) and non-brachycephalic dog owners (NBDOs) in terms of human–dog bonding, behavioral traits, perception of breed-related clinical issues, and their motivations for acquiring their dogs.

## 2. Materials and Methods

### 2.1. Survey Design and Distribution

The same survey design and distribution strategy used in a previous study on brachycephalic cats [20] was applied to explore dog owners’ motivations for acquisition, perceptions of breed-related problems, dog behavior (C-BARQ), human–animal bonding (DORS questionnaire), and the dog’s clinical history. The questionnaire was adapted from a prior survey administered to brachycephalic dog owners in the UK [25,28]. It was distributed to two groups: owners of brachycephalic dogs (BDOs, specifically French and English bulldogs and pugs) and owners of non-brachycephalic dogs (NBDOs, including other non-brachycephalic breeds and mongrels).

The questionnaire consisted of four sections. The first section collected general information on participants (both BDOs and NBDOs), such as age, gender, education, past and current pet ownership, home environment, and other background experiences that could be relevant. The second section gathered demographic information about the dogs (sex, reproductive status, current age, and age at acquisition), origin (e.g., breeder, shelter/rescue, family/friends, or stray), veterinary expenses, expectations, and the perceived level of commitment required for dog care. Owners were also asked to assess their dog’s health status, report major illnesses, and share perceptions about the suffering of brachycephalic breeds. A dedicated sub-section for BDOs included specific questions about the dogs’ medical history and motivations underlying the acquisition of a brachycephalic dog. The third section focused on the human–dog relationship. Owners completed the “Dog–Owner Relationship Scale” (DORS) [29], a component of the Cat–/Dog–Owner Relationship Scale (C/DORS). The DORS is a 28-item Likert-type scale that assesses the quality of pet–owner relationships. It includes both positive items (e.g., “My dog helps me get through tough times”) and negative items (e.g., “There are major aspects of owning a dog that I don’t like”), with responses on a 5-point scale (e.g., “strongly agree” to “strongly disagree” or “very easy” to “very hard”). The C/DORS includes the following three sub-scales:Perceived emotional closeness (PEC, 9 items): this reflects social support, affection, bonding, psychological attachment, companionship, and unconditional love. High scores indicate stronger emotional bonding;The dog–owner interaction (DOI, 10 items): this sub-scale reflects activities related to the physical care of the pet, as well as to more intimate activities, such as kissing, cuddling, and hugging. High scores indicate more frequent and positive interactions;Perceived cost (PC, 9 items): this captures financial, social, and emotional burdens. High scores indicate greater perceived burden and thus a less positive relationship.

Following the authors’ instructions [29], DORS scoring was conducted. Items in the PEC and DOI sub-scales were reverse-scored, so higher values indicate a better perceived relationship. Sub-scale scores were computed for each participant by averaging their item scores, and group means were calculated for BDOs and NBDOs. The reliability of this instrument was good (Cronbach’s alpha = 0.79).

The fourth session included selected items from the Canine Behavioral Assessment and Research Questionnaire (C-BARQ) [30], a widely used and validated tool for evaluating behavior and behavioral problems in dogs. This tool is designed to collect quantitative behavioral evaluations of pets from their owners. In this part of the questionnaire, each behavioral item used a 5-point ordinal rating scale (e.g., never, rarely, sometimes, usually, always) based on observations from the recent past (i.e., in the last few months) [30]. A “not applicable” option was added for those with no experience of the behavior described. For analysis, responses using “not applicable” selections were treated as missing values, with average scores calculated from the remaining responses without missing values. The C-BARQ sections included in the survey were selected because they permitted us to explore the main features that characterize brachycephalic breeds:Trainability (8 items): this evaluates the animal’s disposition to pay attention to its owner, follow simple instructions, learn quickly, retrieve objects, respond positively to correction, and ignore any distracting stimuli.Separation-related behavior (8 items): this assesses signs of anxiety when the dog is separated from its owner (i.e., vocalizing, destructiveness, restlessness, loss of appetite, trembling, and excessive salivation).Excitability (6 items): this measure strong reactions to potentially exciting or arousing events, such as going for walks or car trips, doorbells, the arrival of visitors, the owner arriving home, and the difficulty of settling down after such events.Attachment or attention-seeking (6 items): this evaluates proximity-seeking behaviors, affection solicitation, and agitation when the owner interacts with others.

Higher scores represent more undesirable behavior, except for the trainability sub-scale, where higher scores indicate better performance [30]. The reliability of this instrument was good (Cronbach’s alpha = 0.77).

The questionnaire was in Italian, via the free online platform “Google Forms” (Google). It remained available online from March 2020 until March 2021. A convenience sampling strategy was used by disseminating the survey link through social media platforms (i.e., Facebook and WhatsApp). Participation was restricted to individuals aged 18 or older. Respondents were instructed to answer the questionnaire as fully as possible.

This study was reviewed and approved by the Ethics Committee (reference number 97/20) of the University of Milan, Italy. All participants provided written informed consent prior to participation.

### 2.2. Statistical Analysis

Questionnaire responses were scored and analyzed using IBM SPSS Statistics 29 (SPSS Inc., Chicago, IL, USA). Descriptive statistics were calculated to provide an overview of the two groups of dog owners. Data were tested for normality, and Pearson’s chi-squared test with Bonferroni correction and the Mann–Whitney U test were used to investigate potential group differences in sample characteristics (including the demographics of owners and dogs), owner expectations, veterinarian experiences, and their personal perception of brachycephalic dogs’ health status.

The Mann–Whitney U test was applied to non-normally distributed continuous * categorical data (DORS score * breed and C-BARQ trainability score * breed). The three C-BARQ sub-scale scores (excitability, separation-related behavior, and attachment or attention-seeking) were normally distributed, and a one-way ANOVA test was used to detect possible differences between brachycephalic breeds and non-brachycephalic breeds.

Linear regression was applied to figure out which factor predicted the three C-BARQ sub-scale scores (excitability, separation-related behavior, and attachment or attention-seeking). Sixteen variables were tested for associations with the three C-BARQ sub-scale scores using separate linear regression models: dog demographics (breed, age, sex, age at acquisition, and source); owner demographics (sex and family members); veterinary routines; owner expectations of the breed versus the reality of ownership (veterinary costs, activity level, and overall behavior); and owner perceptions of brachycephalic health.

The DORS data were not normally distributed. A Kruskal–Wallis equality of population rank test was employed to determine if the DORS score was significantly associated with dog characteristics, such as breed, age, sex, age at acquisition, and source, the dog’s clinical data, owner demographics (gender and family members); veterinary routines; owner expectations of the breed versus the reality of ownership (veterinary expenses, activity level, and overall behavioral traits).

To investigate possible differences in C-BARQ and DORS sub-scale scores among the three brachycephalic breeds (pug, French bulldog, and English bulldog), a Kruskal–Wallis test was used.

## 3. Results

### 3.1. Section 1: Demographic Information

A total of 728 respondents completed the survey (*n* = 320 for brachycephalic dog owners (BDOs); *n* = 408 for non-brachycephalic dog owners (NBDOs)). Of the overall sample, 88,6% (645/728) were female. The largest age group was 31–44 years, accounting for 38.5% (280/728) of participants. Most respondents (48.8%, 355/728) held an undergraduate degree as their highest level of education. Regarding living conditions, 53% (386/728) lived in a house with a garden, 35.9% (261/728) lived in an apartment with a balcony, and only 11.1% (81/728) lived in an apartment without outdoor access. In terms of household composition, 40% (291/728) lived in a two-person household and 52.7% (384/728) reported having children in the household. A total of 78.3% of respondents had previously owned a dog (570/728). Statistically significant differences between the BDO and NBDO groups were observed in relation to age, living environment, education level, and previous dog ownership. The detailed demographic characteristics of the two groups are summarized in Table 1.

### 3.2. Section 2: Dog Demographics, Clinical History, and Owner Expectations

One-third of the dogs (33.9%, 247/728) were between 1 and 3 years of age. In the study population, 48.9% (356/728) of the dogs were female, of which 25.1% were intact, while 51.1% (372/728) were male, of which 44.2% were intact. Within the non-brachycephalic group, 15.1% (62/408) were mixed-breed dogs, while the most frequently represented pure breeds were Golden retrievers and Labrador retrievers (12.2%, 50/408). Among the brachycephalic dogs, the distribution by breed was as follows: 38.1% (124/320) French bulldogs, 33.1% (106/320) English bulldogs, and 28.1% (90/320) pugs.

Most dogs were acquired between two and four months of age (74.6%, 543/728), and 45.7% were acquired from pedigree breeders. Breed-specific demographic data are presented in Table 2.

The largest percentage of dogs (44.4%, 323/728) underwent a veterinary check-up once per year. A total of 19% of dogs (139/728) had never received a veterinary check-up, while 6.9% (50/728) were taken to the veterinarian more than five times per year.

Brachycephalic breeds were believed by most owners to suffer more than non-brachycephalic ones (79.3%, 578/728); 82.1% (598/728) of owners perceived these breeds as having more health problems compared to other dogs. In terms of behavior, 59.9% of owners (436/728) reported that their dog’s interactions with other dogs were normal. Overall, most owners (both BDOs and NBDOs) reported that their expectations were met across various aspects of dog ownership: veterinarian expenses (50.8% (370/728)); activity level (62.5% (455/728)); interactions (52.9% (385/728)); and general behavior (41.7% (304/728)). Table 3 shows the differences between the two groups. The frequency of veterinary checks, interactions with other dogs, satisfaction with the activity level of their dog, satisfaction with the interactions, and satisfaction with the general behavior were statistically significant.

#### Brachycephalic Dog Owners

A total of 77.5% (248/320) of owners were aware of breathing disorders (BOAS) prior to acquiring their dog, and 77% (240/320) were also aware of other breed-specific health disorders (i.e., ocular, skin, heart, neurological, and reproductive diseases). When asked to evaluate their dog’s health status, 36.9% (118/320) rated it as *excellent* and 44.1% (141/320) as *good*, while just 4.7% (15/320) and 1.6% (5/320) rated it as *poor* and *very bad*, respectively. The predominant factors influencing owners’ decisions to acquire these breeds were behavioral traits (94.1%, 301/320) and aesthetic appeal (74.4%, 238/320). Regarding specific physical traits, 84.4% (270/320) of owners loved the large, round eyes, 81.6% (261/320) favored the rounded head, and 95.3% (305/320) expressed a preference for the flat face. The most loved traits included cuteness (92.8%, 297/320), an appearance that evoked protectiveness (64.7%, 207/320), and a human-like face (88.8%, 284/320). The dogs’ health information reported by BDOs is summarized in Table 4.

### 3.3. Section 3: DORS Scale

DORS scores were high across all three sub-scales, with both means and medians exceeding 4.0 for dog–owner interaction (DOI) and perceived emotional closeness (PEC). The perceived cost (PC) sub-scale score was significantly higher in dogs living in houses compared to those living in apartments, while the DOI sub-scale score was higher in dogs living in apartments compared to those living in houses (*p* ≤ 0.05) (Figure 1 and Figure 2).

The mean PC sub-scale score tended to increase with the age of the dog, whereas the DOI sub-scale score increased up to approximately one year of age, then tended to decrease as the dog aged (*p* ≤ 0.05) (Figure 3 and Figure 4).

The DOI sub-scale score was significantly higher in intact male dogs (*p* ≤ 0.05) (Figure 5). The PEC sub-scale score was higher in dogs adopted before two months of age, whereas the DOI sub-scale score was higher in dogs adopted between two and four months (*p* ≤ 0.05) (Figure 6 and Figure 7). The lowest PEC and DOI sub-scale scores were reported for stray dogs (*p* ≤ 0.05). Mean DOI and PEC scores were significantly higher in BDOs compared to NBDOs (*p* ≤ 0.05), whereas the mean PC score was higher in NBDOs than in BDOs (Table 5).

When analyzing only brachycephalic breeds, the PEC sub-scale score was significantly higher among owners who chose these breeds based on their character (*p* ≤ 0.05). The PC and DOI sub-scales scores were associated with the perceived health status of the dog: a higher PC score was reported by owners who rated their dog’s health as “excellent”, while a higher DOI score was associated with dogs rated as having “poor” health (*p* ≤ 0.05). Additionally, the DOI sub-scale score was higher in brachycephalic dogs without respiratory symptoms and in those not showing symptoms while sleeping (*p* ≤ 0.05).

### 3.4. Section 4: C-BARQ

The mean C-BARQ scores were significantly higher in the non-brachycephalic group compared to the brachycephalic group for trainability (*p* < 0.001), excitability (*p* = 0.003), and separation-related behaviors (*p* = 0.024) (Figure 8, Figure 9 and Figure 10). No statistical differences were found between the groups for attachment/attention-seeking.

Breed type (brachycephalic versus non-brachycephalic) was significantly associated with the excitability, separation-related behavior, and attachment/attention-seeking sub-scale scores in the linear regression models (*p* ≤ 0.05). Owner and dog age, as well as owner expectations regarding the level of interaction with the breed, were associated with all three sub-scales (*p* ≤ 0.05). Expectations regarding behavior and the source of dog acquisition were associated with the separation-related behavior and attachment/attention-seeking sub-scales (*p* ≤ 0.05). Age at acquisition and veterinary routine were significantly associated only with the attachment/attention-seeking sub-scale (*p* ≤ 0.05). The overall regression models were statistically significant for all three sub-scales: the model for excitability showed an *R*^2^ of 0.58 (*F* = 4.856, *p* < 0.001); the model for separation-related behaviors had an *R*^2^ of 0.88 (*F* = 7.656, *p* < 0.001); and the model for attachment/attention-seeking reported an *R*^2^ of 0.91 (*F* = 7.952, *p* < 0.001). When comparing the brachycephalic breeds, pugs scored highest in all three sub-scales: excitability (*p* = 0.007), separation-related behaviors (*p* = 0.005), and attachment/attention-seeking (*p* = 0.011). French bulldogs showed intermediate scores, while English bulldogs had the lowest scores in all three sub-scales (Figure 11, Figure 12 and Figure 13).

When analyzing only brachycephalic breeds and their health information, higher scores in the separation-related behavior and attachment/attention-seeking sub-scales were significantly associated with dogs reported to have ocular symptoms (*p* ≤ 0.05).

## 4. Discussion

This study provided a snapshot of various aspects related to brachycephalic dog ownership in Italy, offering specific insights into owners’ experiences and expectations and perceived health status, human–dog bonding, and behavioral traits in brachycephalic breeds such as the French bulldog, English bulldog, and pug. As previously described, brachycephalic dog ownership is a phenomenon that continues to fuel the so-called *brachycephalic boom* [1,28]. Over the years, these dogs have been selectively bred to accentuate extreme morphological features in response to increasing demand from owners drawn to neotenous morphological traits, such as a very flat face [1,18,19]. This selective breeding has had detrimental consequences for animal health and welfare, leading to the development of various breed-related disorders, including BOAS, ocular conditions, and gastrointestinal issues [3,5,7,12,15]. Despite the well-documented health concerns, their popularity worldwide, including in Italy, remains high. As confirmed by the findings of this study, the French bulldog is currently the most popular brachycephalic breed [28].

Further investigation into the motivations and perceptions of individuals who choose to own brachycephalic dogs, as well as a deeper understanding of the human–dog bond and the behavioral traits of these breeds, is essential for addressing the ongoing popularity of these animals and the associated welfare concerns. Based on the results of this study, the topical profile of a brachycephalic dog owner in Italy is a woman aged between 31 and 54, living in an apartment (with or without outdoor access), and with previous dog ownership experience. Women, once again, were more likely to respond to the survey and appeared to be more actively involved in pet management and care, as also observed in prior research [31,32]. Most households consisted of two people, and 60.3% included children. These findings are not totally in line with previous studies, which reported that brachycephalic dog owners (BDOs) tend to be younger than non-brachycephalic dog owners (NBDOs), typically aged 25–34 years [25,33]. However, in line with the existing literature, the majority of participants in this study were female, and a high proportion of brachycephalic-owning households included children [25]. Even though there are many common aspects, this point may be a distinction between brachycephalic dog ownership and brachycephalic cat ownership. In a similar study conducted on brachycephalic cat breeds, most cat-owning households did not include children [20]. Being child-safe was the most important behavioral trait when owners were asked to describe the ideal companion dog [34,35]. Although there is no scientific evidence suggesting that brachycephalic dog breeds pose a lower bite risk to children, owners may perceive them as more suitable due to their gentle appearance and presumed behavior [27,36]. This perception is considered one of the relevant factors fuelling brachycephalic popularity. Another widespread belief among owners is that brachycephalic dogs are better suited to the demands of a frenetic and modern lifestyle. The owners perceive these breeds as requiring less exercise or space, making them more suitable for apartment living and for modern lifestyles [1,27,36]. In addition, these breeds are frequently believed to be “lazy”. This is inherently biased. The main motivation for exercise reduction in brachycephalic breeds can be linked to respiratory difficulties (i.e., BOAS), which owners frequently fail to recognize as a clinical problem [3,12,15,21,25]. The notion that brachycephalic breeds align well with owners’ lifestyles is supported by findings from this study. Most BDOs reported satisfaction with their dogs’ activity levels, interactions, and veterinary costs, suggesting that ownership met their expectations. This is consistent with previous research showing that while many owners found veterinary costs and exercise demands to be as expected, approximately one-fifth had underestimated them [25]. Our findings are further supported by results from the DORS scale, which showed a significantly higher score for the perceived cost (PC) sub-scale among NBDOs compared to BDOs. This sub-scale captures the negative aspects of pet ownership, including financial, social, and emotional burdens. Higher PC scores indicate that these aspects negatively impact the owner and suggest a relationship of lower overall quality. In this regard, it was demonstrated that owners of chronically or terminally ill pets may experience a greater caregiver burden that encompasses emotional, physical, social, and financial impacts [16,37]. Despite this, BDOs in the present study appeared to be more satisfied with this aspect: the fatigue of owning their own dog is considered high, but it meets expectations. It is plausible to assume that BDOs anticipate elevated veterinary expenses and do not perceive them as excessive or overwhelming [1,25]. This can also point out fallacies in owners’ perceptions of clinical issues that are considered “normal” for the breed. The onset of health problems in brachycephalic dogs appears to be anticipated by many owners and may be considered as part of their commitment to the breed [21,28,38]. This is substantiated by the fact that the majority of BDOs were aware, prior to acquiring their dog, of specific breed-related health issues in the overall population of brachycephalic dogs (i.e., BOAS or ocular and skin disorders). This awareness contrasts with findings from similar studies on brachycephalic cat ownership, where most owners were unaware of breed-related health problems [20]. Paradoxically, almost all BDOs in this study agreed that brachycephalic breeds suffer more and have more health issues than others. In line with this, common clinical signs such as regurgitation after or during meals, snoring during sleeping, and breathing distress after activity or in hot climatic conditions were frequently observed by BDOs. In addition, BOAS and gastrointestinal disorders were among the most commonly diagnosed conditions [6,7,8,28]. Additionally, ocular and dermatological surgeries were the most frequent conformational interventions required for these breeds [8,10,11,13]. Despite this, most owners rated their dog’s health status as good or excellent when asked to assess it. All of these conflicting results support the normalization phenomenon, in which BDOs may consciously recognize the presence of health issues but unconsciously accept them as typical or even inevitable characteristics of the breed [5,23,25]. This paradox is further highlighted by the analysis of the correlation between dog–owner relationship quality and health status and clinical problems. Within the BDO group, a rating of “excellent” health was linked to higher scores in the perceived cost (PC) sub-scale, while higher dog–owner interaction (DOI) scores were associated with the absence of respiratory symptoms and sleep-related issues. This latter result can have a logical explanation since the absence of clinical problems can increase the quality of the relationship. However, the finding that high scores also correlated with “poor” health ratings suggests a more complex dynamic. One possible explanation is that poor health and the associated caregiving responsibilities may fulfill a psychological need in some owners, reinforcing the bond through caregiving behaviors [1,25].

This interpretation aligns with the well-known psychological mechanism defined as cognitive dissonance. Cognitive dissonance is the discomfort a person feels when his/her behavior does not align with his/her values or beliefs. It is a psychological phenomenon that occurs when a person holds two contradictory beliefs at the same time. Because the dissonance is aversive, the individuals try to reduce it by changing one of the other beliefs [39]. In the specific case of brachycephalic breeds, owners are aware of the health issues in their dog’s breed but find it emotionally difficult to acknowledge them in their own dog. Instead, they divert the problems to other individuals as a strategy for coping with negative emotions. In other words, it is common for BDOs to believe that their dogs are healthier than average for their breed, recognizing the breed-related problems but normalizing and accepting them [23,25]. Furthermore, the strong emotional bond with brachycephalic dogs can result in the minimization or denial of health problems, amplifying the effect of cognitive dissonance [27,40,41].

The human–dog relationship has been shown to be influenced by the morphological characteristics of brachycephalic breeds, particularly their facial appearance of a large head, a round face, a high and protruding forehead, large eyes, bulging cheeks, and a small nose and mouth. These features are referred to as the “baby schema” described in Lorenz’s theory. According to this theory, there is an innate release mechanism for caregiving and affective orientation towards individuals who exhibit these morphological traits, typical of infants, which also stimulates human nurturing behaviors [42,43,44]. Neotenic features are attractive. Brachycephalic dogs retain infant traits into adulthood and are considered cute for this reason [1,19]. Cute dogs are perceived to be more amicable, affectionate, and with more desirable behavioral traits [18,22]. Indeed, positive behavioral traits are deemed to be typical attributes of brachycephalic breeds. Exploring this aspect in the current study, satisfaction regarding the behavior of their own dog was better than expected in most BDOs (52.5%). Conversely, only a small proportion of BDOs (4.7%) stated that their expectations were not totally met, as shown in a previous study [25]. Brachycephalic breeds are generally perceived as good companion dogs, characterized by positive behavioral traits or good temperaments. This perception is also one of the main motivations for acquiring these breeds, as confirmed by the present study, followed closely by their aesthetic appearance [23,25,27,28,36]. A high PEC score that was correlated with character as a motivation for purchasing a brachycephalic dog further supports this finding. Previous research has shown that brachycephalic breeds displayed eye contact more quickly with unfamiliar people and followed pointing gestures more readily than non-brachycephalic breeds. They tend to pay more attention to human faces and gestures, and for these reasons, they are often considered affectionate and collaborative companions [45,46,47]. Additionally, during specific behavioral tests, it was observed that brachycephalic dogs looked longer at human and dog portraits than non-brachycephalic breeds, suggesting morphological influences on social interactions [45,48]. It is possible that the behavioral traits are a result of the anatomical structure of the dogs’ eyes, as their visual acuity is higher in the center of the visual field and lower at the periphery, allowing them to focus more on their communication partner, such as their owner [47,49]. In addition, the neotenic facial features of these dogs may encourage humans to engage in mutual gazing, providing brachycephalic dogs with more opportunities to acquire skills in paying attention to humans and making eye contact with them [50]. The ability to form eye contact with humans is a trait highly valued by owners, and it can be one explanation for the great popularity of these breeds [1,49], as frequent eye contact is known to strengthen the dog–owner bond [51]. These features of brachycephalic breeds also support the idea that they are more trainable and attached to their owners. However, the results of the present study did not confirm this assumption: non-brachycephalic dogs scored significantly higher in trainability according to the C-BARQ assessment. This finding supports the results found by Brincat et al. (2021) [52], who reported that brachycephalic dogs obtained the lowest score for trainability in the C-BARQ protocol. Mesocephalic breeds were the most trainable, but both brachy- and dolichocephalic breeds fell into the “less trainable” category, as previously observed by Helton (2009) [53]. In our study, attachment levels did not show statistical differences between the two groups. If considering only the brachycephalic group, attachment, as well as separation anxiety, were associated with the presence of ocular symptoms. Evidence suggests that owners of animals exhibiting clinical disorders tend to develop stronger attachment bonds, likely due to the increased caregiving and protective behaviors elicited by the animals’ perceived vulnerability [23,33,54]. The selection of neotenic physical traits has likely been accompanied by the selection of neotenous behaviors, such as attachment and behavioral dependency [47]. For this reason, brachycephalic breeds could be more prone to attachment and separation anxiety-related disorders [55,56,57]. Comparing the different brachycephalic breeds involved in this study, pugs showed higher scores than other breeds for excitability, separation-related behaviors, and attachment/attention-seeking. This may be linked to the extremely exaggerated brachycephalic morphology typical of this breed.

It is evident that owner interpretations of canine behavior, as well as the perception of relationship quality, can be strongly influenced by the dog’s neotenic appearance [18,28,38,52,57,58]. Further investigation is needed to determine whether less extreme brachycephalic conformations might influence these perceptions. Understanding which physical traits should no longer be selected for is essential to improving the health and welfare of these animals, while still preserving the “cuteness” that attracts owners to purchase these breeds [27,59].

This study has some limitations. First, the use of a convenience sample limits the generalizability of the findings to the broader Italian dog-owning population. As has already been observed in the literature, most participants were women [20,31,32]. To reduce gender-related sampling bias, future studies should attempt to recruit more balanced sampling, including more male respondents. Additionally, the self-selection of participants introduced a systematic bias, as the sample was not randomly selected, which may have affected the representativeness of the results [60]. In addition, because the survey was administered online, only respondents who had Internet access and a certain level of digital literacy could participate. Being self-reported, the information collected may have been influenced by how participants interpreted the questions. In addition, several owners may have been hesitant to confess their pet’s clinical issues or their own emotions and perceptions regarding brachycephalic dog ownership [61]. Regarding health status, this aspect was explored only within the brachycephalic group, which represents an additional limitation of this study. Furthermore, the data reflect a specific moment in time, namely during COVID-19 restrictions in Italy, which may have influenced respondents’ experiences and behaviors [62,63]. This study is descriptive and correlational in nature, aiming to explore multiple dimensions of brachycephalic dog ownership. Despite its limitations, it offers valuable insight into the complex and multifaceted nature of the brachycephalic dog ownership phenomenon.

## 5. Conclusions

The relationship between humans and brachycephalic dogs is widely recognized as complex and multifaceted. Italian owners’ beliefs about these breeds and their motivations for purchasing them (including perceived positive behavioral traits, good temperament, and appealing appearance) are consistent with findings reported in the literature. Despite growing awareness of health and welfare issues, the popularity and demand for brachycephalic dogs, especially subjects with extreme traits, persist. The under-recognition or minimization of welfare and health problems in these breeds helps owners cope with the psychological discomfort of having chosen a dog with inherent health issues. Expanding scientific research on different aspects of the brachycephalic phenomenon, considering possible differences between brachycephalic dog and cat ownership, can be relevant to understanding the different facets of this complex issue. A deeper understanding can support the development of effective strategies to protect the welfare and health of brachycephalic breeds. Scientific knowledge should be leveraged to revise breed standards, prioritizing animal welfare while still maintaining their “cuteness”. Ethical breeding practices are urgently needed and cannot be delayed further. To find a resolution to the ongoing brachycephalic crisis, this approach could be more effective than banning these breeds.

## Figures and Tables

**Figure 1 animals-15-01496-f001:**
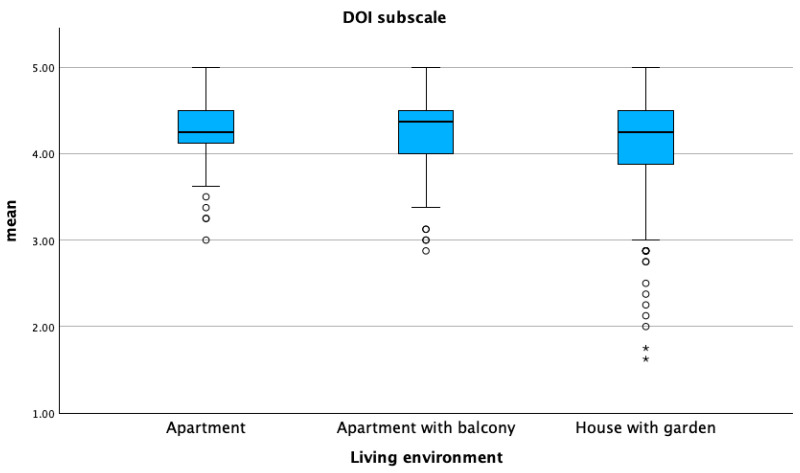
Mean distribution of the dog–owner interaction sub-scale (DOI) with respect to the living environment of dogs. Dots represent weak outliers; stars represent strong outliers.

**Figure 2 animals-15-01496-f002:**
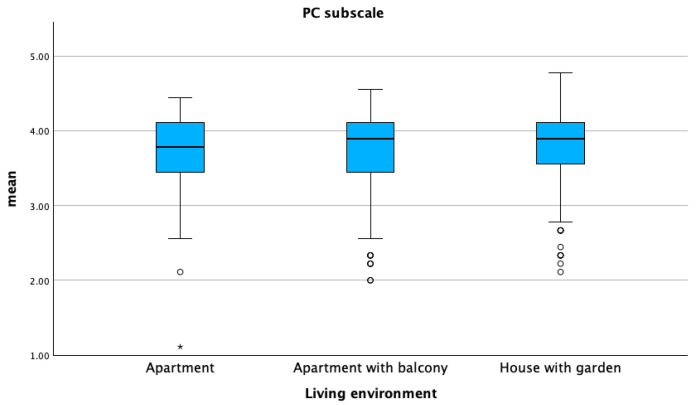
Mean distribution of perceived cost sub-scale (PC) with respect to the living environment of dogs. Dots represent weak outliers; stars represent strong outliers.

**Figure 3 animals-15-01496-f003:**
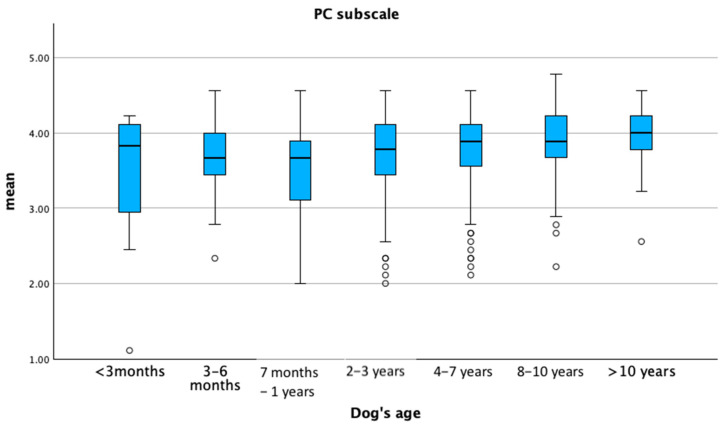
Mean distribution of the perceived cost sub-scale (PC) according to the dog’s age. Dots represent weak outliers.

**Figure 4 animals-15-01496-f004:**
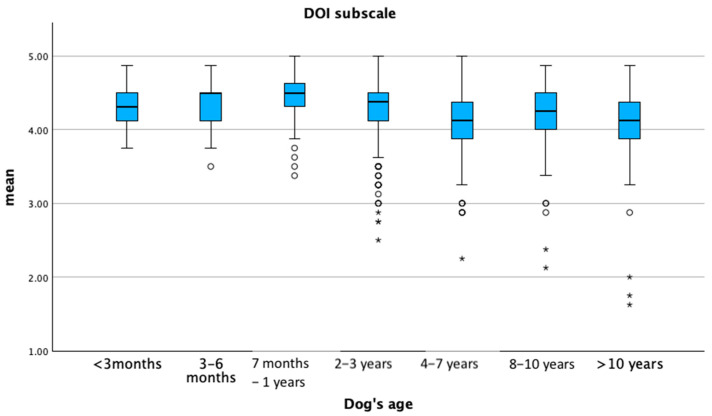
Mean distribution of the dog–owner interaction sub-scale (DOI) according to the dog’s age. Dots represent weak outliers; stars represent strong outliers.

**Figure 5 animals-15-01496-f005:**
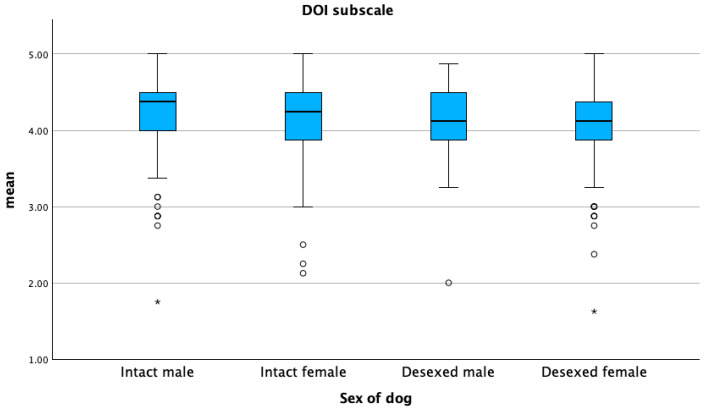
Mean distribution of the dog–owner interaction sub-scale (DOI) according to the sex of the dog. Dots represent weak outliers; stars represent strong outliers.

**Figure 6 animals-15-01496-f006:**
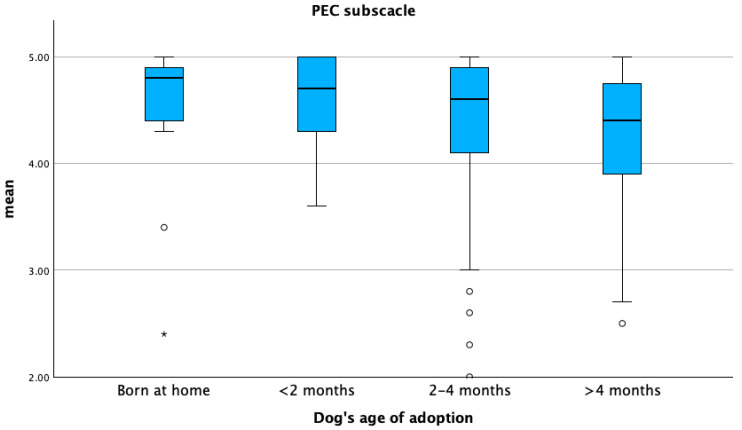
Mean distribution of perceived emotional closeness (PEC) with respect to the age at acquisition of the dog. Dots represent weak outliers; stars represent strong outliers.

**Figure 7 animals-15-01496-f007:**
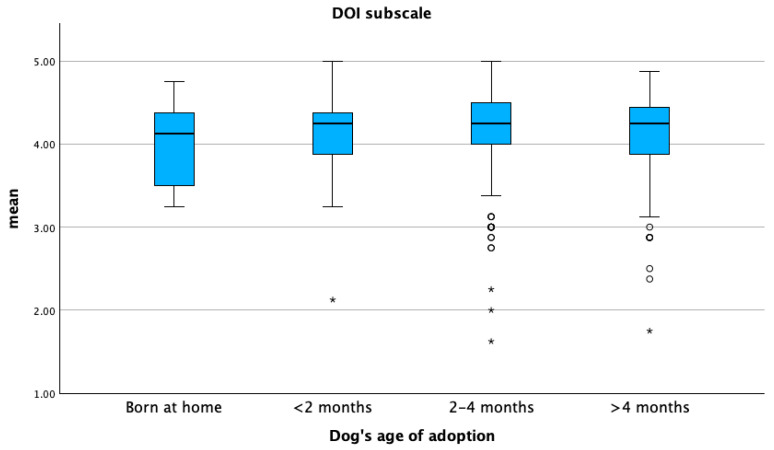
Mean distribution of the dog–owner interaction sub-scale (DOI) with respect to the age at acquisition of the dog. Dots represent weak outliers; stars represent strong outliers.

**Figure 8 animals-15-01496-f008:**
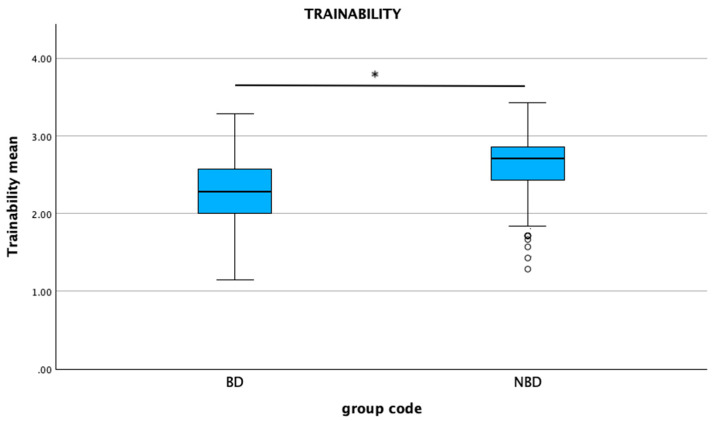
Comparison of mean C-BARQ scores for trainability between brachycephalic and non-brachycephalic breeds. * *p* < 0.001. Dots represent the outliners.

**Figure 9 animals-15-01496-f009:**
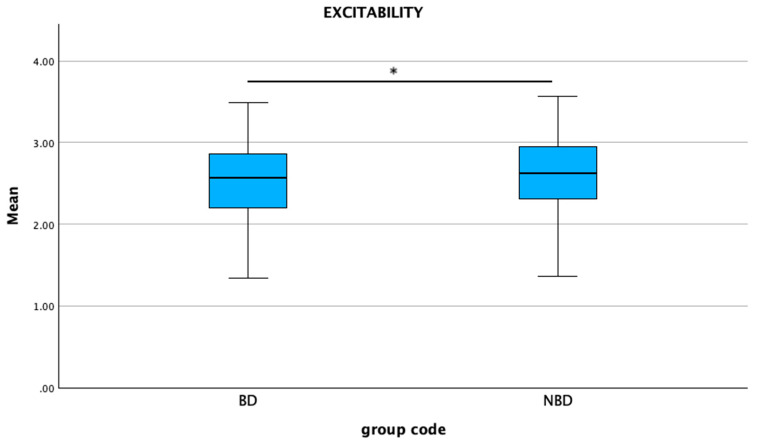
Comparison of mean C-BARQ scores for excitability between brachycephalic and non-brachycephalic breeds. * *p* = 0.003.

**Figure 10 animals-15-01496-f010:**
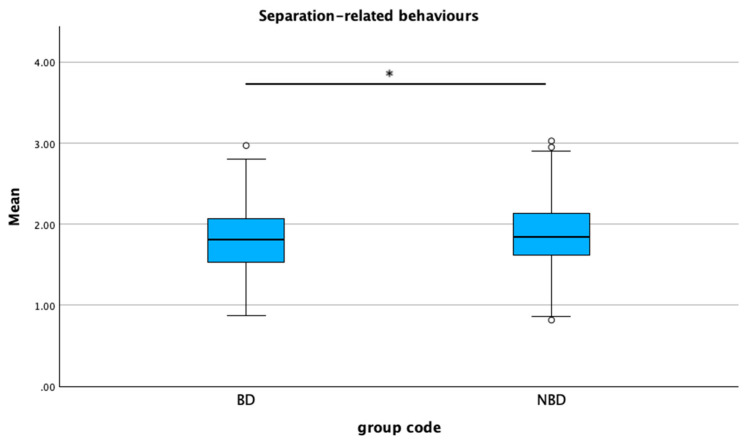
Comparison of mean C-BARQ scores for separation-related behaviors between brachycephalic and non-brachycephalic breeds. * *p* = 0.024. The dots represent the outliners.

**Figure 11 animals-15-01496-f011:**
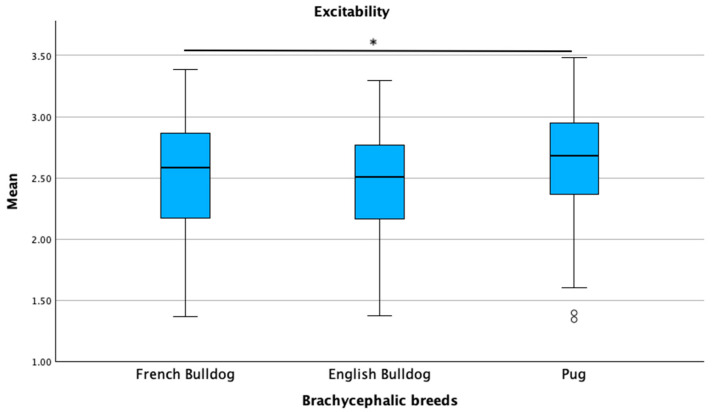
Comparison of mean C-BARQ scores for excitability among French bulldogs, English bulldogs, and pugs. * *p* = 0.007.

**Figure 12 animals-15-01496-f012:**
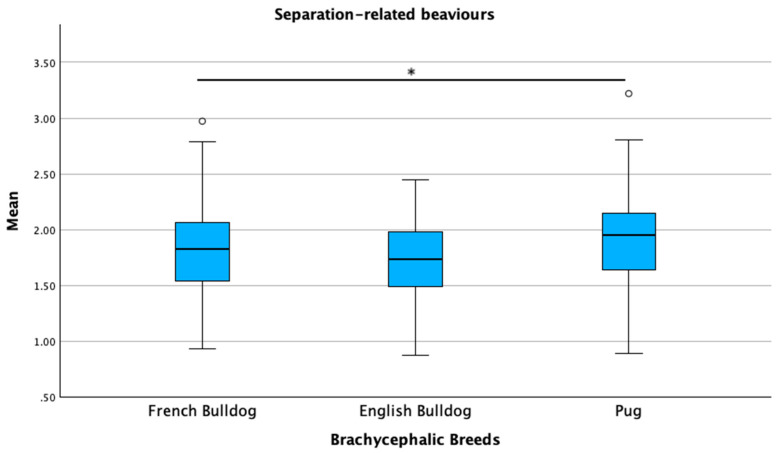
Comparison of mean C-BARQ scores for separation-related behaviors among French bulldogs, English bulldogs, and pugs. * *p* = 0.005. Dots represent outliers.

**Figure 13 animals-15-01496-f013:**
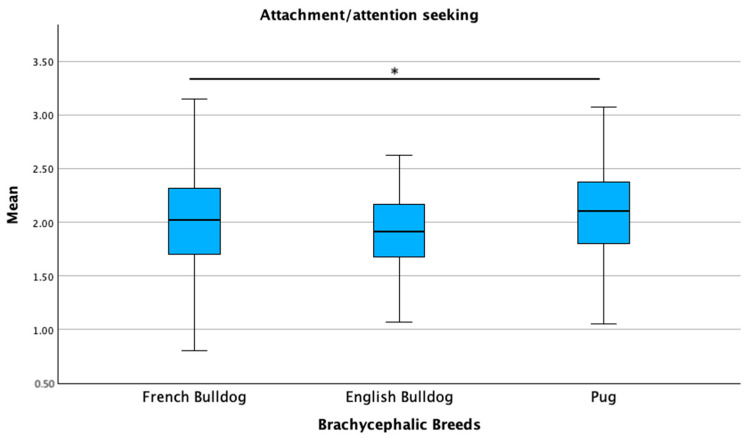
Comparison of mean C-BARQ scores for attachment/attention-seeking among French bulldogs, English bulldogs, and pugs. * *p* = 0.011.

**Table 1 animals-15-01496-t001:** Representation of owners’ demographic aspects by group (NBDOs: non-brachycephalic dog owners; BDOs: brachycephalic dog owners).

		NBDOs		BDOs	
		N°	%	N°	%
Sex	Female	365	89.5%	280	87.5%
	Male	38	9.3%	35	10.9%
	I prefer not to specify	5	1.2%	5	1.6%
Age *	18–30 years	149	36.5%	66	20.6%
	31–44 years	153	37.5%	127	39.7%
	45–54 years	64	15.7%	100	31.3%
	55–64 years	37	9.1%	21	6.6%
	65–74 years	3	0.7%	3	0.9%
	>75 years	2	0.5%	3	0.9%
Environment *	Apartment	118	28.9%	143	44.7%
	apartment with a balcony	249	61.0%	137	42.8%
	semi-detached or detached house	103	41.2%	91	32.9%
Education level *	middle or secondary school	212	52%	206	64.4%
	undergraduate or post-graduate degree	196	48%	114	35.6%
People in the household	1	48	11.8%	35	10.9%
	2	170	41.7%	121	37.8%
	3	100	24.5%	86	26.9%
	4	73	17.9%	58	18.1%
	5	11	2.7%	16	5%
	>5	6	1.5%	4	1.3
Children in the household	No	217	53.9%	127	39.7%
	Yes	191	46.1%	193	60.3%
Previously * owned a dog	No	68	16.7%	90	28.1%
	Yes	340	83.3%	230	71.9%

* Significant differences are marked with an asterisk (*p* ≤ 0.05).

**Table 2 animals-15-01496-t002:** Representation of dog demographics by breed.

		NBDOs		BDOs	
		N°	%	N°	%
Age *	Less than 3 months	4	1%	4	1.3%
	3–6 months	10	2.5%	23	7.2%
	7 months–1 year	13	3.2%	34	10.6%
	2–3 years	131	32.1%	116	36.3%
	4–7 years	120	29.4%	87	27.2%
	8–10 years	74	18.1%	35	10.9%
	More than 10 years	56	13.7%	21	6.6%
Sex *	Intact male	161	39.5%	161	50.3%
	Intact female	114	26.9%	69	21.6%
	Desexed male	35	8.6%	15	4.7.%
	Desexed female	98	24%	75	23.4%
Age at acquisition *	Born at home	11	2.7%	2	0.6%
	Less than 2 months	27	6.6%	14	4.4%
	2–4 months	291	71.3%	252	78.8%
	More than 4 months	79	19.4%	52	16.3%
Source *	Certified breeder (recognized by the Italian Authority)	196	48%	137	42.8%
	Non-certified breeder	22	5.4%	48	15%
	Private/born at home	106	26.0%	104	32.5%
	Rescue	67	16.4%	13	4.1%
	Stray	15	3.7%	3	0.9%
	Other (Internet, shops, etc.)	2	0.5%	15	4.7%

* Significant differences are marked with an asterisk (*p* ≤ 0.05).

**Table 3 animals-15-01496-t003:** Differences in responses between NBDOs and BDOs (NBDOs: non-brachycephalic dog owners; BDOs: brachycephalic dog owners).

		NBDOs		BDOs	
		N°	%	N°	%
Veterinary check-ups *	never	74	18.1%	65	20.3%
	1 time per year	241	59.1%	82	25.6%
	2–3 times for year	69	16.9%	85	26.6%
	4–5 times per year	19	4.7%	43	13.4%
	> 5 times per year	5	1.2%	45	14.9%
Brachycephalic breeds suffer more than others	Yes	315	77.2%	263	82.2%
	No	93	22.8%	57	17.8%
Brachycephalic breeds have more problems than others	Yes	329	80.6%	269	84.1%
	No	79	19.4%	51	15.9%
When your dog interacts with other dogs, what generally happens? *	Interact normally	246	60.3%	190	59.4%
	Ignore each other	47	11.5%	42	13.1%
	The other dog attacks mine	15	3.7	40	12.5%
	My dog attacks the other dog	46	11.3	29	9.1%
	Other	54	13.2%	19	5.9%
Satisfaction with veterinary expenses	Less than expected	84	20.6%	73	23.0%
	Meets expectations	214	52.5%	156	49.1%
	More than expected	110	27.0%	89	28.0%
Satisfaction with the activity level of your dog *	Less than expected	58	14.2%	51	16.0%
	Meets expectations	242	59.3%	213	66.8.0%
	More than expected	108	26.5%	55	17.2.0%
Satisfaction with interactions (play, cuddle requests, etc.) *	Less than expected	46	11.3%	15	4.7%
	Meets expectations	220	53.9%	165	51.9%
	More than expected	142	34.8%	138	43.4%
Satisfaction with general behavior *	Better than expected	152	37.3%	168	52.5%
	Meets expectations	177	43.4%	127	39.7%
	Worse than expected	79	19.4%	25	7.8%

* Significant differences are marked with an asterisk (*p* ≤ 0.05).

**Table 4 animals-15-01496-t004:** Dogs’ health information reported by BDOs.

		BDOs	
		N°	%
Symptoms during and after meals	Vomiting	36	11.3%
	Choking	26	8.1%
	Difficulty breathing	39	12.2%
	Regurgitation	63	19.7%
Symptoms during sleeping	Snoring	200	62.3%
	Changing positions	98	30.6%
	Chin in an elevated position	65	20.3%
	Open mouth breathing	43	13.4%
	Apnea	25	7.8%
Other symptoms	Epiphora	88	27.5%
	Noisy breathing	123	38.4%
	Breathing distress after activity	206	64.4%
	Coughing	28	8.8%
	Breathing distress in hot climatic conditions	182	56.9%
	Heat intolerance	6	1.9%
Veterinary diagnosis	BOAS	64	20%
	Laryngeal Collapse	31	9.7%
	Entropion	34	10.6%
	Ectropion	19	6.9%
	Gastrointestinal disorders	72	22.5%
Conformation-related surgeries	Ocular surgery	31	9.7%
	Corrective surgery (cutaneous and respiratory)	28	8.8%
	Odontostomatologic surgery	12	3.8%

**Table 5 animals-15-01496-t005:** Mean scores of the three DORS sub-scales in the brachycephalic dogs (BDs) and non-brachycephalic dogs (NBDs).

Sub-Scale	NBDOs	BDOs
DOI (Dog–Owner Interaction)	4.08 ± 0.52	4.32 ± 0.4
PEC (Perceived Emotional Closeness)	4.33 ± 0.56	4.55 ± 0.44
PC (Perceived Cost)	3.84 ± 0.44	3.68 ± 0.56

## Data Availability

The datasets generated and analyzed during the current study are available from the corresponding author upon reasonable request, as they are not publicly accessible due to privacy considerations.

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
