# Peer review of "Italians Can Resist Everything, Except Flat-Faced Dogs!â€"

_animals, 2025, doi:10.3390/ani15101496_

Round 1
Reviewer 1 Report
Comments and Suggestions for Authors
Thank you for this survey paper. A few comments:
- Title: I dislike commenting on titles, however this one was hard for me to reconcile. If the largest part of your sample are NBDOs, and a minority are BDOs, can we say that Italians cannot resist BDOs if the larger part of your sample are NBDOs?
- There is inconsistency in your acronyms. Please choose either NBDO/BDO or NBCO/BCO.
- Can you give more support/explanation for why you chose the specific sections of the CBARQ over others?
- Line 226 - can you specify that they were adopted between 2-4 MONTHS OF AGE? There could be other interpretations of the "2-4 months" other than age.
- Line 230 - It may be clearer to use "largest percentage of dogs". "Most of dogs" to me should read as a percentage larger than half of the sample.
- Table 4 could have benefitted from a comparative sample from the NBDOs. As these questions were not provided to NBDOs, it may be worth noting in the limitations that there is no comparison group for this section of the survey.
- Figures 1-7 could benefit from changing the chart titles to the full words rather than the acronym. This will make it easier to consume for readers.
- In the abstract (and later in the discussion), it is stated that "Most BDOs were aware of the incidence of these disorders in bracycephalic dogs." What I was unable to find in the text was whether BDOs were aware of the incidence in the overall population and whether they are aware of the incidence in their own dogs (i.e., did they think their dog was unaffected or the exception compared to the larger population)?
This manuscript could benefit from review for grammar by someone with mastery of the English language (US).
Author Response
Dear reviewer,
We appreciated your suggestions and comments to improve our manuscript. We addressed all of them and we have made a careful revision of English with the help of a colleague with mastery of the English language .
Title: I dislike commenting on titles, however this one was hard for me to reconcile. If the largest part of your sample are NBDOs, and a minority are BDOs, can we say that Italians cannot resist BDOs if the larger part of your sample are NBDOs?
Thanks for pointing this out. The title doesn’t refer to the numerosity of the sample but it refers to the factors and the psychological mechanisms that influence the brachycephalic owners’ decision-making. We have chosen this title as we believe it can be appropriate, as it catches the attention in a nice and funny way, paraphrasing a famous aphorism of Oscar Wild.
There is inconsistency in your acronyms. Please choose either NBDO/BDO or NBCO/BCO.
Sorry we corrected it. It was a typo
Can you give more support/explanation for why you chose the specific sections of the CBARQ over others?
Thanks for pointing this out. We decided to select the following sections: trainability, separation related, excitability, attachment because these behavioural traits can be overlapped to the more searched features that characterize the brachycephalic breeds. Moreover, we had to reduce the CBARQ questionnaire because to long for our survey. We added this in the materials and methods (LL 139-140).
Line 226 - can you specify that they were adopted between 2-4 MONTHS OF AGE? There could be other interpretations of the "2-4 months" other than age.
We specified LL 221
Line 230 - It may be clearer to use "largest percentage of dogs". "Most of dogs" to me should read as a percentage larger than half of the sample.
We modified the text LL 226
Table 4 could have benefitted from a comparative sample from the NBDOs. As these questions were not provided to NBDOs, it may be worth noting in the limitations that there is no comparison group for this section of the survey.
Thank you for this comment. We agree and we added this as a limitation of the study. LL 527-528
Figures 1-7 could benefit from changing the chart titles to the full words rather than the acronym. This will make it easier to consume for readers.
Thanks for pointing this out. The acronym was explained in the caption so we would prefer to keep the title compact in order to maintain lightweight the image and to avoid redundancy
In the abstract (and later in the discussion), it is stated that "Most BDOs were aware of the incidence of these disorders in bracycephalic dogs." What I was unable to find in the text was whether BDOs were aware of the incidence in the overall population and whether they are aware of the incidence in their own dogs (i.e., did they think their dog was unaffected or the exception compared to the larger population)?
Thanks for that comment. We specified that Most BDOs were aware of the incidence of these disorders in the overall population of brachycephalic dogs. LL 417-419
Reviewer 2 Report
Comments and Suggestions for Authors
Hi,
Thank you so much for the opportunity to review this paper. I thought it was extremely interesting to get further insight of what might be internal human motivations for getting brachycephalic dogs despite the external seemingly apparent issues in their ownership. I have provided some minor comments and suggested revisions below
Line 166:
Was there a minimum amount of time the dog and owner had to have lived together as inclusion/exclusion criteria?
Line 225:
Potential error of just “French Bulldog” in this line of text
Line 227:
It might be helpful for context what is the process for being certified by Italy as a breeder and how that differs from non-certified breeder. Would a non-certified breeder be akin to someone like the colloquial term “backyard breeder” in the US?
Table 2,Lines 226:
Just a matter of preference but I know for most readers in the US, the use of “adopted” and “adoption” really relates to acquiring a dog from a rescue rather than any form of acquisition. It might be better to frame it more broadly as plainly “source” and “age at acquisition” or something similar.
Line 240:
When you describe this as “statistically significant” is this just omnibus or is there post-hoc testing to show where the significant difference lies, if there were post-hoc analysis, it might be key to indicate those differences with asterisks in the table to greater clarity.
Line 395:
I am a little confused about the sentence “This is an inherently bias.” Do you mean “This is inherently biased”?
Line 404, 411, 413, 420:
Believe “BCOs” should be “BDOs” for text consistency.
Line 411:
Confused by line “The fatigue of owning the own dog..”
Line 498: I generally shy away from “proven” since that is a very loaded term, might be better to say “it has been shown..”
Author Response
Hi,
Thank you so much for the opportunity to review this paper. I thought it was extremely interesting to get further insight of what might be internal human motivations for getting brachycephalic dogs despite the external seemingly apparent issues in their ownership. I have provided some minor comments and suggested revisions below
Dear reviewer,
We really appreciate all the comments and suggestions, and the work done to improve our paper. We tried to address all of them as appropriately as possible.
Line 166:
Was there a minimum amount of time the dog and owner had to have lived together as inclusion/exclusion criteria?
Thank you for pointing this out. For this survey we did not consider this factor. We agree that this point could be interesting to explore better in the future.
Line 225:
Potential error of just “French Bulldog” in this line of text
We deleted the typo
Line 227:
It might be helpful for context what is the process for being certified by Italy as a breeder and how that differs from non-certified breeder. Would a non-certified breeder be akin to someone like the colloquial term “backyard breeder” in the US?
Yes, an Italian non-certified breeder can be defined as backyard breeder of US. In the text we added a quick definition of certified breeders (certified breeder is legally and professionally recognized by an Italian public body). In table 2 we had already specified it.
Table 2, Lines 226:
Just a matter of preference but I know for most readers in the US, the use of “adopted” and “adoption” really relates to acquiring a dog from a rescue rather than any form of acquisition. It
might be better to frame it more broadly as plainly “source” and “age at acquisition” or something similar.
Ok we modified them across the text
Line 240:
When you describe this as “statistically significant” is this just omnibus or is there post-hoc testing to show where the significant difference lies, if there were post-hoc analysis, it might be key to indicate those differences with asterisks in the table to greater clarity.
We used SPSS to analyzed the data. We performed Bonferroni as post-hoc but this test in SPSS didn’t show where the significant difference lies. So, “the statistically significant” is just omnibus
Line 395:
I am a little confused about the sentence “This is an inherently bias.” Do you mean “This is inherently biased”?
We modified accordingly LL 395-396
Line 404, 411, 413, 420:
Believe “BCOs” should be “BDOs” for text consistency.
We modified accordingly
Line 411:
Confused by line “The fatigue of owning the own dog.”
We meant that despite the difficulty of caring for a brachycephalic dog, it meets the expectations of its owners.
Line 498: I generally shy away from “proven” since that is a very loaded term, might be better to say “it has been shown.”
We agree and we modified the sentence accordingly LL 498-501
Reviewer 3 Report
Comments and Suggestions for Authors
This study demonstrates that despite the inherent health issues associated with brachycephaly in dogs, Italian owners are still highly motivated to own them. This study therefore replicates and builds upon previous findings in other human populations, suggesting that the issue is ubiquitous amongst brachycephalic dog owners.
The manuscript was a pleasure to read, with the introduction nicely setting the context for the study.
The methods clearly describe the process and the statistical analysis is appropriate. I would however like to see the Cronbach’s alpha for this population for both DORS and C-BARQ added to the methods.
The results are also easy to follow and the tables are very clear. A few specific comments:
Line 225: There appears to be an additional ‘French Bulldog’ typed after the full stop.
Line 241: Think you could delete ‘resulted’ here.
The discussion nicely interprets and places the results in the context of previous results and was also a pleasure to read.
Line 457: Think this should read “baby schema” not scheme.
Comments on the Quality of English LanguageA minor amount of English language checking is needed throughout – for example the figure legends for figs 1 & 2.
Author Response
This study demonstrates that despite the inherent health issues associated with brachycephaly in dogs, Italian owners are still highly motivated to own them. This study therefore replicates and builds upon previous findings in other human populations, suggesting that the issue is ubiquitous amongst brachycephalic dog owners.
The manuscript was a pleasure to read, with the introduction nicely setting the context for the study.
Dear reviewer,
We are delighted to receive this positive feedback of our work. We would like to thank you for the work done to improve our manuscript. We also accepted and addressed all the comments and suggestions.
The methods clearly describe the process and the statistical analysis is appropriate. I would however like to see the Cronbach’s alpha for this population for both DORS and C-BARQ added to the methods.
Thank you very much for pointing this out. We added the information as requested in Methods
Cronbach’s alpha: 0.79 DORS LL 128-129
Cronbach’s alpha: 0.77 CBARQ LL 154-155
The results are also easy to follow and the tables are very clear. A few specific comments:
Line 225: There appears to be an additional ‘French Bulldog’ typed after the full stop.
Thanks for pointing that out. We deleted it
Line 241: Think you could delete ‘resulted’ here.
We modified the text accordingly
The discussion nicely interprets and places the results in the context of previous results and was also a pleasure to read.
Line 457: Think this should read “baby schema” not scheme.
Thank you for pointing that out. It was a typo, we corrected it accordingl